# The Impact of Campus Outdoor Space Features on Students’ Emotions Based on the Emotion Map

**DOI:** 10.3390/ijerph20054277

**Published:** 2023-02-28

**Authors:** Jun Zhang, Yiran Li

**Affiliations:** School of Landscape Architecture, Northeast Forestry University, Harbin 150040, China

**Keywords:** mood map, campus external space, spatial features, facial expression recognition, emotion

## Abstract

To explore the influence of campus public space characteristics on students’ emotions, we investigated the association mechanism between public space characteristics and students’ emotions concerning the distribution of students’ emotions in public spaces. The present study used photographs of facial expressions taken over two consecutive weeks as a source of data regarding the students’ affective reactions. The collected facial expression images were analyzed using facial expression recognition. Values were assigned to the expression data, combined with geographic coordinates to create an emotion map of the campus public space using GIS software. Then, spatial feature data via emotion marker points were collected. We used smart wearable devices to combine the ECG data with spatial characteristics and took SDNN and RMSSD as ECG indicators to assess mood changes. We analyzed the correlation between these spatial features and heart rate variability and developed regression models for the ECG data. The findings show that sky visibility, space D/H, green visibility, skyline change index, and boundary permeability can engage students’ positive emotions in a meaningful way. On the other hand, paving visibility and road linearity tends to induce negative emotions in students’ minds.

## 1. Introduction

Positive psychology suggests that emotional states are closely linked to physical and mental health and social adjustment [1], and that facial reactions are an important conduit for outward internal emotions [2]. Human emotions, on the other hand, do not change spontaneously but emerge through environmental or event stimuli and are subject to some variability in individual behavior, physiological mechanisms, and cognitive appraisals [3]. Differences in emotional responses elicited by different spatial environments and their methods of measurement have thus been a central research question in the field of spatial planning. Of these, students’ mental health and the impact of the college campus space on students’ emotions have been topics of intense interest in recent years [4]. From a developmental psychology perspective, college students are approaching the end of their transition from the campus environment to the social environment and from the student environment to adulthood (typically 18–27 years of age) [5]. During this time, college students are required to complete several tasks such as living independently from family care, setting self-development goals, attending to interpersonal relationships, and adapting quickly to a series of changes in their new lifestyles, study content, developmental goals, interpersonal relationships, and roles to prepare for their transition into society and their families. Sudden changes and stress can easily lead to bad feelings such as jealousy, depression, anxiety, fear, and low self-esteem, which have a significant impact on college students’ physical and mental health [6,7]. Over the past few years, social and academic circles have begun to pay a great deal of attention to college students’ health and other health problems [8,9]. Environmental factors can affect students’ psychology, behavior, health, mood, and so on. Over the last 30 years, researchers from abroad have demonstrated the health-promoting effects of the natural environment and elucidated the mechanisms of influence on the natural environment in the fields of anthropology [10,11,12,13], health medicine [14], psychology [15,16], epidemiology [17,18], and design [19,20,21]. One such widely acknowledged theory is the Stress Recovery Theory (SRT) [22], which proposes that visual perception of nature can reduce levels of stress, and another is the Attention Restoration Theory (ART) [23], which proposes that a beautiful natural environment can elicit an undirected focus of attention to alleviate directed attentional fatigue. Other scholars have summarized nature’s contribution to health in four major ways: “repairing stress and mental damage”, “building social cohesion”, “improving air quality”, and “promoting physical activity” stimulating physical activity [24] [25]. Over the past decade, domestic scholars have begun to pay more and more attention to environmental health benefits [26], rehabilitation landscape design [27], evidence-based design [28,29], and the role of rehabilitation garden research in old-age institutions have become more and more important [30,31]. The role of the natural environment in promoting health has been widely recognized in academia.

As a very important public space in the city, the campus space of colleges and universities, as a full-time place of higher education, differs from primary and secondary schools in that students have longer and more frequent contact with the campus space of colleges and universities [32]. Particularly in recent years when the epidemic was recurrent and schools have been under closed control, college campuses take over students’ daily living, study, and rest needs and provide them with more serial sound conditions; the leisure and stress-relieving role of the campus environment is therefore highlighted [33]. As the era continues to develop, there is also a need to improve the design of outdoor space on campus. The current design of campus spaces emphasizes good indoor space and lacks attention to the environmental design of outdoor public spaces on campus [34]. The design of the campus’s external spatial environment must be continually improved as it is a necessary place for students’ daily lives, study, and communication. However, most universities’ current planning and design of campus spaces cannot meet the health needs of students [35,36] because the design of the campus is primarily fast-paced, neglecting to consider students’ health needs. To pursue the shortest construction period, most campus spaces are built with “shape over efficiency” and “quantity over quality”, and students’ mental stress is not effectively relieved [37,38].

The only important medium for school students to be exposed to the natural environment during the epidemic was the outdoor public space on campus, and mood changes were an important indicator for observing mental health. With campus outdoor public space characteristics and emotions as the primary research variables, the purpose of this paper is to address the following questions: (1) Does the campus outdoor public space environment have a moderating effect on emotions? (2) What are the factors that characterize campus outdoor public space that may alter college students’ emotions? (3) Given the answers to the above two questions, can suggestions be made for the promotion of students’ mental health in the planning and design of the space on college campuses?

## 2. Materials and Methods

### 2.1. Selection of Study Sites and Subjects

The Northeast Forestry University has a campus in Harbin, the center of China’s largest state forested area, with an area of 126.6247° E to the east and 45.7662° N. The campus has an area of 136 hectares. The Northeast Forestry University campus is divided into two parts, the old campus, and the new campus, and the activity spaces within the campus cater to the needs of the students for a variety of activities. Based on existing studies, four types of campus activity spaces are extracted, which are the living area, teaching area, recreation area, and sports area [38], and a total of eight typical activity spaces are selected from four types of partitions. Figure 1 shows the selected partition.

### 2.2. Survey Content and Method

Psychologist Mehrabian’s research shows that emotional expression consists of 7% speech, 38% voice, and only 55% facial expressions [39]. Meng Zhaolan, our psychologist, has shown that facial expressions can be used as an objective cue for emotional research [40]. Furthermore, live captured facial expressions are more realistic than web-crawled images [41], so live-captured images are taken as the source of emotional data. Emotional characteristics were investigated over two time periods, from 8:00 a.m. to 11:30 a.m. and from 2:30 p.m. to 5:30 p.m. Participants were asked to rate their emotional characteristics. Studies have shown that the collection of facial expressions for two consecutive weeks can accurately express the distribution of emotional features across the region [42]. Additional filtering and cropping of the images of the students’ facial expressions collected during the study were necessary to meet the recognition requirements (images required a clear face, goggles, and mask to ensure that 3/4 of the face was exposed, and fewer than 3/4 of the face images were null and void; the selection of null images was eliminated) [43]. The photographs collected met facial recognition criteria and were spontaneous or recreational, thereby reducing the number of bystander-type images (staying in each activity area for more than 5 min). The number of samples needed for space was calculated based on Edward’s interpersonal spatial distance of 1.0 to 1.3 m and 3.5 m^2^ per person [44]. Images of facial expressions collected over two consecutive weeks in different sample spaces were collated, and the GPS coordinates corresponding to the images were extracted with the aid of LocaSpaceViewer, resulting in a collection of sentiment units and GPS coordinates for matching sample spaces. This resulted in a total of 7092 images, 5402 of which were identifiable by face choice.

### 2.3. Survey Content and Method

Research in cognitive psychology has shown that emotional arousal can be better measured separately from emotional efficacy to obtain data on changes in the mood [45]. In this paper, we focus on how campus spatial factors reduce students’ negative emotions and promote students’ positive emotions. This study does not specifically investigate the influence of campus space factors on specific emotions but does analyze the relationship between positive and negative emotions and the campus space. Ekman’s classification study divides personal emotions into seven categories, namely stillness, joy, concern, sadness, anger, hurt, and shock [46]. Therefore, based on Ivan et al.’s analysis of the characteristics of spatial emotional types and classification of emotions, emotions are divided into positive, negative, and natural emotions. Natural emotions (neutral) were scored as 0, positive emotions (happy, surprised) as 1, and negative emotions (fear, sadness, disgust, anger) as −1. We use the arithmetic mean of students’ spatial expressions as a measure of the spatial features of emotional distribution [47].

Based on richness, abundance, accessibility, science, and the reliability of data acquisition and processing, analysis of the collected images was performed using a facial expression recognition system (FaceReader 9.0), which has an 89% self-test recognition accuracy rate. FaceReader 9.0 is a software package by VicarVision and Noldus Information Technology based on the theory of automated analysis of facial expressions and facial motion coding (FACS) systems [48]. FaceReader can capture the intensity of an image’s expression, type of emotion, validity, and arousal through the process of face recognition, in addition to the approximate age and gender of the object presented in the image (shown in Figure 2).

### 2.4. Emotion Metric and Emotion Map Construction

Emotion validity and arousal are obtained through emotion metrics, and emotion attributions are made to form a coordinate set of the emotion data in each space relative to the extracted GPS coordinates. The site study used CAD to draw each spatial plane and GIS software to draw the heat map of the emotional state. We combined the traced floor plan with the affective heat map to produce a distribution map of affective features across each sample space (as shown in Figure 3). Emotion marker points were selected from each space based on the emotion map search: the positive emotion-gathering point, the negative emotion-gathering point, and the neutral emotion-gathering point. Spatial feature data were collected at the emotion label points. Figure 4 shows the mood map and selected points from each public space.

### 2.5. Evaluation of Spatial Characteristics

Quantitative index selection of elements of spatial characteristics: Based on the comprehensive study of the environmental characteristics’ assessment of campus public space, through on-site research on campus public space, the campus public space index system affecting students’ emotions was broken down into three first-level indices: spatial attribute elements, elements of physical microenvironmental features, and elements of landscape facilities. Because the experiment is based on visual perception, the spatial features that can be perceived through visual perception are selected as the subjects of the experiment. Twelve spatial feature elements were selected: skyline change index (SCI), space D/H (D/H), sky visibility (SV), paving visibility (PV), road linearity (RL), green visibility (GV), plant type number (PTN), number of signs (NUOSI), number of facilities type (NFT), number of seats (NUOSE), boundary permeability (BP), and site height differences (SHD). The distance based on human eye recognition in the study data is 250 m, and the distance beyond the horizon has not been calculated. Prior quantification of spatial characteristics such as green visibility, sky visibility, and paving visibility was typically carried out by taking photographs and then using the gridding method, i.e., by manually determining the proportion of vegetated area in an image within a grid cell. Quantifying these landscape features does not reflect the true level of greenness visible in the space surrounding the human eye since most quantification data are measured as a function of field of view extent. This paper uses a spherical panoramic image, rather than the two-dimensional one used in conventional vision.

### 2.6. Research Methodology

Twenty-four urban and rural planning and landscape planning students were chosen as subjects for this study, and no heavy exercise was required before the experiment (avoiding excessive heart rate after exercise). As the physical micro spatial elements can also affect students’ changes in mood, to avoid the influence of physical micro spatial elements such as sound, light, heat, and wind environment, this study was carried out in a single season and experimental conditions such as wearing headphones and shadowing were selected. Surveys were conducted between 15 July and 26 July 2022 during sunny mornings from 9:00 to 11:30 and afternoons from 15:00 to 17:30, during a season of lush vegetative growth and sunny and mild weather. There were two parts to the formal survey: ECG data collection and a mood state survey. Each participant was assigned a smart ECG device that could connect to a smartphone and transmit the uploaded data in real-time via Bluetooth. Subjects had a mean age of 23 ± 4 years, with 12 males and 12 females. Participants were asked to follow a predetermined path to a predetermined 24 experimental sites. Participants were instructed to take 5 min to calm down when they arrived at each scheduled location on campus. At this point, the smart wearable began recording ECG data for over 5 min, at the end of which they were asked to complete questionnaires to report their emotional state. To provide a complete assessment of mood changes, this study used a combination of subjective questionnaires and objective physiological indicators, which yielded more standardized data [49]. Participants were required to perceive for five minutes at a predetermined location [50], yielding the participant’s ECG data for those five minutes. We then took pictures of the visual environment at that time and collected data on the spatial features that required on-site research. We used these methods to collect spatial feature data, ECG data, and a few subject-level samples over the course of the experiment.

### 2.7. Emotional State Assessment

The BFS Mindfulness Scale (Befindlichkeitsskalen), a German-language instrument for the measurement and description of mental state, was used in the subject selection, and it is a better reflection of the relationship between the state of mind and subjects’ emotions. Developed by Abele and Brehm in 1986 [51], it is a commonly used instrument in current research in German-speaking countries. Each of the subscales of the BFS Mood Scale consists of five questions, each of which is an adjective that describes a state of mind, corresponding to an individual’s psychological feelings, with a total of 40 questions, scored in 5-point Likert segments, and all questions are mixed and randomized.

The scale was translated into Chinese by Dr. Gang-Yan Si in 1995 and was subsequently tested for reliability and validity and showed that the BFS has good theoretical and structural validity [52].

National and international studies have shown that positive emotions can uncouple and restore the activation state of a variety of stress-induced cardiovascular activities and quickly return them to normal levels [53]. For this reason, the Positive and Negative Affect Scale, PANAS, was used as a measure of mood change in the test phase of the investigation of spatial features and physiological changes. The PANAS scale, developed by Watson D et al. [54], is a state-based mood scale, which is primarily influenced by situational factors and was the Chinese version of the PANAS was piloted by Professor Yang Tingzhong et al. and is suitable for use in China for our study [55].

### 2.8. Selection of Physiological Variables

Individual heart rate variability is affected to some degree by emotion, and the effect of different emotions on individual heart rate variability varies. Electrocardiogram (ECG) testing is now widely used as an important method of determining changes in mood state [56]. Numerous studies have shown that negative emotions can make individuals prone to loss of control over their behavior and more prone to impulsive anger [57]. Heart rate variability is divided into time-domain and frequency-domain indicators, and time-domain indicators are often chosen to be the standard deviation of the RR interval (SDNN (ms)); root means square of adjacent normal RR interval difference (RMSSD (ms)); percentage of adjacent normal RR interval difference over 50 ms (PNN50); triangular index; maximal heart rate; minimal heart rate; and average heart rate [58]. In this case, the SDNN standard deviation and the root mean square of the difference between adjacent normal RR intervals (RMSSD (ms)) were chosen to represent mood change.

## 3. Results and Analysis

### 3.1. Analysis Method

Variance and correlation tests as well as multiple regression analysis were performed on campus outdoor spatial feature data and experimental samples using the SPSS software [59].

The Kolmogorov—Smirnov test was initially used to analyze the normality of the data collected. As can be seen, the *p*-value of the data samples collected was greater than 0.05, and the difference was statistically significant. Second, the Pearson correlation was used to compute the relationship between campus spatial features and neural networks, and SDNN and RMSSD responses as well as t-tests (*p* < 0.01 and *p* < 0.05) were used to test for significant differences. Third, the regression equations between the spatial characteristics of the college campuses and the SDNN and RMSSD neural networks were established by linear regression. The significance of the regression equations was examined by ANOVA [60]. Figure 5 shows the entire process.

This section may be divided into subheadings. It should provide a concise and precise description of the experimental results, their interpretation, as well as the experimental conclusions that can be drawn.

### 3.2. Analysis Method

The descriptive statistics of the collected physiological data showed that SDNN and RMSSD both had higher positive emotionality than negative emotionality, in agreement with the results of Wang Shuangxi et al. [59] as seen in Table 1.

The Pearson correlations between the external spatial characteristics of the college campus and the SDNN and RMSSD neural networks can be seen in Table 2 and Table 3.

The results showed that Pearson correlation coefficients between seven spatial features including sky visibility, boundary permeability, space D/H, skyline change index, paving visibility, green visibility, and road linearity showed significance. Among them, sky visibility, boundary permeability, space, skyline change index, and green visibility are significantly and positively correlated with the SDNN The results showed that there was a significant correlation between the correlation coefficient and the Pearson correlation coefficient. These values have a significant and negative correlation with paving visibility and road linearity. Road linearity and boundary permeability *p* > 0.01 (correlation coefficients of 0.419 and 0.477, respectively) were less correlated with the other spatial features, and there were no significant correlations between the plant type number, number of seats, number of signs, facilities of type, site height differences, and SDNN (*p* > 0.05).

In RMSSD, there was a significant positive correlation between sky visibility, boundary permeability, space D/H, and green visibility, as well as a significant negative correlation between paving visibility and road linearity. Of these, boundary permeability (BP) and green visibility *p* > 0.01 (correlation coefficients of 0.485 and 0.477, respectively) were found to be less relevant than the rest of the spatial features. There was no significant effect of plant type number, number of signs, number of seats, number of facilities type, and site height differences on RMSSD (*p* > 0.05) There were no significant main effects of plant type.

In general, the elements of spatial attributes dominate the influence on SDNN and RMSSD, among which sky visibility, space D/H, skyline change index, paving visibility, and road linearity have been shown to have a significant influence on mood. As for the landscape facility elements, the correlations between the plant type number, the number of signs, the number of seats, the number of facility types, and the site height differences and SDNN and RMSSD were not significant. This indicates that these characteristics have a weak effect on mood. Of these, the effect of the green-view rate on SDNN and RMSSD was large, which initially indicated a positive effect on mood with an increasing rate of green gaze.

### 3.3. Analysis Method

For spatial features that were significantly correlated with heart rate variability, additional regressions and analyses were performed based on linear regression models of spatial features with SDNN and RMSSD (as shown in Table 4 and Table 5). There were significant effects of sky visibility, paving visibility, space D/H, and the skyline change index on both SDNN and RMSSD (*p* < 0.01), and the degree of the effect was large (R^2^ > 0.6). For example, green visibility, road linearity, and boundary permeability had a smaller effect on SDNN and RMSSD (R^2^ < 0.5), whereas road linearity, boundary permeability, and green visibility had influence coefficients of 0.151, 0.625, and 0.087 for SDNN, respectively, with *p* values greater than 0.05 for the RMSSD model.

Table 1 shows that each spatial feature has an effect on the two HRV indicators for assessing change in mental stress (R^2^ > 60); the two models passed the F test with test coefficients of 130.679 and 122.949, respectively, and the variance inflation factor (VIF < 5) makes the multiple linear regression model more reliable as well.

## 4. Discussion

The purpose of this study was to combine heart rate variability data acquired using a smart wearable device with the spatial characteristics of a college campus. The use of portable heart rate recording instruments was convenient for the field test and reduced the psychological burden of the testers. In terms of spatial feature extraction, the panoramic method was used for the calculation of greenness, openness, and the hardening degree to avoid the subjectivity and uncertainty of data obtained by traditional calculation methods that rely on indicators such as photo angle and lens focal length. Concerning the selection of mood indicators, previous studies have found SDNN and RMSSD heart rate variability to be physiologically objective indicators of subjects’ changes in mood, and the selection of objective physiological indicators in real-time also avoids some of the drawbacks of using subjective questionnaires alone. One of the most important contributions of this study is the use of the mood map approach to test site selection, which provides a novel and more objective and accurate method for exploring the effects of public space characteristics on mood change on university campuses.

This study has some shortcomings, however; no study has confirmed that SDNN and RMSSD heart rate variability can be defined as either positive or negative mood states when they reach a certain threshold, but only as a proxy for the pattern of mood change itself. Due to the management of epidemic closure, only one school was selected for the trial sample and sample selection may have limitations, thus future trials with multiple schools may be feasible.

## 5. Conclusions

Based on an emotion map, this study combines campus spatial characteristics with physiological indicators and found that different types of spatial characteristics have different effects on human physiological indicators. Correlation analysis and multiple linear regression analysis allowed us to clarify the influence relationship between the ECG indicators and the different spatial features:

(1)Indicators of campus spatial features are related to public space type, and indicators of the same features often vary greatly across different spatial types. This is a side indication of how much students like different public spaces. The combination of the emotion map constructed in this study demonstrates that emotions in the leisure and sports space tend to be positive. In contrast, emotions in the teaching and living space tend to be more negative.(2)Of the effects of different spatial characteristics of the campus on students’ physiological changes, for the spatial attribute items, the six different spatial characteristics were all significantly correlated with heart rate variability. Of these, sky visibility, the skyline change index, the paving visibility, and space D/H have a stronger impact on physiological indicators, such that they have a greater impact on the students’ emotions. In terms of the landscape attribute items, only green visibility was significantly correlated with heart rate distortion, whereas the other items did not correlate significantly with heart rate variability. There was a positive correlation between sky visibility, skyline change index, boundary permeability, space D/H, green visibility, and heart rate variability. There was a significant and negative correlation between the paving visibility and road linearity and physiological indicators. For example, in the case of the positive sentiment point, the mean sky visibility was 33.54%, the space D/H value was 4.03, and the RMSSD and SDNN values were 46.88 ms and 76.01 ms, respectively. The mean sky visibility for the negative sentiment point was 26.88%, the space D/H value was 2.43, and the RMSSD and SDNN values were 35.12 ms and 58.59 ms, respectively. SDNN and RMSSD heart rate variability has been shown to correlate positively with mood changes [59]. We can see that by increasing sky visibility, the skyline change index, boundary permeability, space D/H, and green visibility, there is a positive impact on the student’s mood. As the paving visibility and road linearity increases, it can be detrimental to student mood.(3)This study finally attempts to construct a multiple linear regression model with campus spatial characteristics and teleological indicators as variables, making it possible to rationally design indicators of spatial characteristics based on the need to positively influence students’ emotions. This indicates that we can control spatial features based on physiological indicators in the design of the campus space, which has some practical importance for building a healthy campus. The method is, however, only an attempt in the present study, and further research is required to confirm this.

In this study, a campus spatial emotion map was constructed using the facial expression recognition method, which combined the spatial features of a college campus with physiological indicators and extended the applicability of environmental psychology and a healthy environment to some extent. In terms of the empirical data, the purpose of this study was to quantify indicators of campus spatial characteristics through a physiological experiment, which confirmed from an objective perspective that spatial characteristics have significant effects on students’ mood changes. However, shortcomings remain in this study as the sample space was selected from a single school during the experiment due to the pandemic, and there were wide differences in indicators of spatial characteristics of different types of campus space. Furthermore, because standard physiological indicator values vary between individuals and the sample size is relatively small, we were not able to accurately explore different spatial features up to a certain level to get the best value for positively influencing students’ emotions, which we believe should be the primary direction for future research on object-level spatial features.

## 6. Optimization Suggestions

Using the public campus space of Northeast Forestry University as a case in point, the results of this paper confirm that there is a correlation between students’ emotions and the public space of college and university campuses. Furthermore, it confirms that there is variability in the influence of different factors, and that elements of the campus public space influence students’ affective changes to varying degrees. Despite the clear and significant influence of spatial attributes on students’ emotions, although the influence of landscape installations on students’ emotions is relatively small, landscape installations also have some influence on students’ emotions. We thus propose three spatial emotion guidelines in combination with prior studies conducted by different control items.

On the one hand, a positive affective tone is set by the elements of spatial ownership in terms of spatial interface form. The continuous building interface should not be too long, and the layout of the interface building close to the main road should be less continuous. The teaching space and D/H living space should be controlled at 1 to 2 [61], and recreational and sporting space should be greater than 4 whenever possible with a recommended value of 8. Pedestrian roads should avoid short-distance zigzagging, and the closest principle should be adopted for the passing road as much as possible. The zigzagging region is expected to increase the path length to improve the coupling. Plaza space should control the proportion of hard pavement, the degree of hardening should make the hard pavement < 30%, and large-area pavement should be avoided to reduce the monotony of the space.

Again, students’ emotional motivation is promoted by adjusting aspects of the natural classroom landscape element. Primarily green visibility, plant species, and site height difference are three factors to be regulated. In terms of visual perception, it is recommended that the green space view rate should be within 40–70% [62]. The green coverage of the landscape within the range of visual perception can be appropriately increased within a reasonable range with enriching plant types, optimizing plant species’ configuration, and selecting appropriate tree species. For campus spaces with large height differences, the design of the site should conform to the topographical contours and protect the ecology of the original site.

Lastly, negativity is avoided by adjusting the elementary aspects of the service facilities. There is a need for appropriate increases in the types and numbers of fitness, recreational, and rest facilities in the space. The types and number of other facilities in the space would need to be reduced. Attention needs to be given to collocation and the arrangement of various kinds of facilities; resting and service facilities should be placed at the edge of the space as much as possible. Facilities such as bulletin boards and trash receptacles may be placed at the entrances and exits of the space or on the side of the boundary road; placing them within the public space should be avoided as much as possible.

## Figures and Tables

**Figure 1 ijerph-20-04277-f001:**
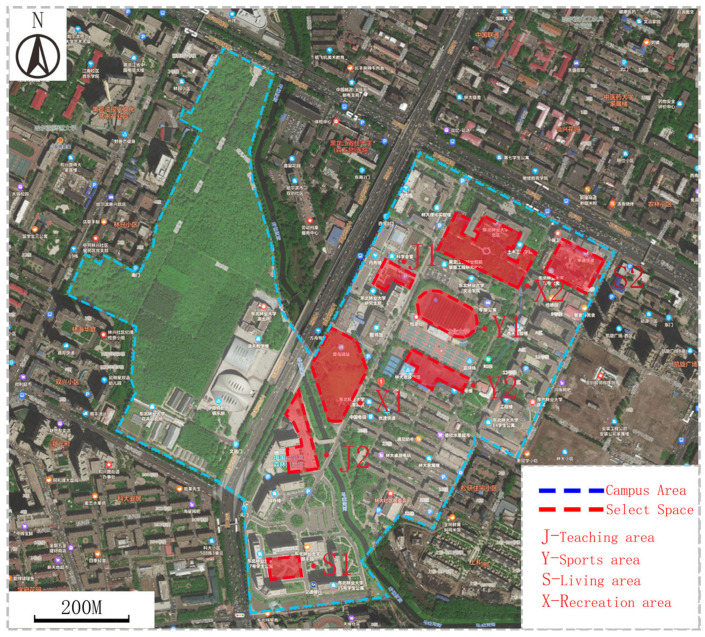
Diagram the selection of public space on campus. (Diagram: Sample spatial distribution map).

**Figure 2 ijerph-20-04277-f002:**
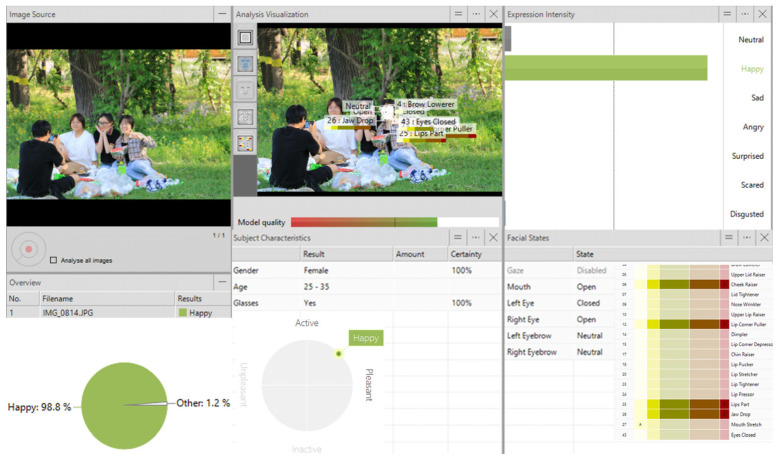
FaceReader 9.0 Facial Analysis.

**Figure 3 ijerph-20-04277-f003:**
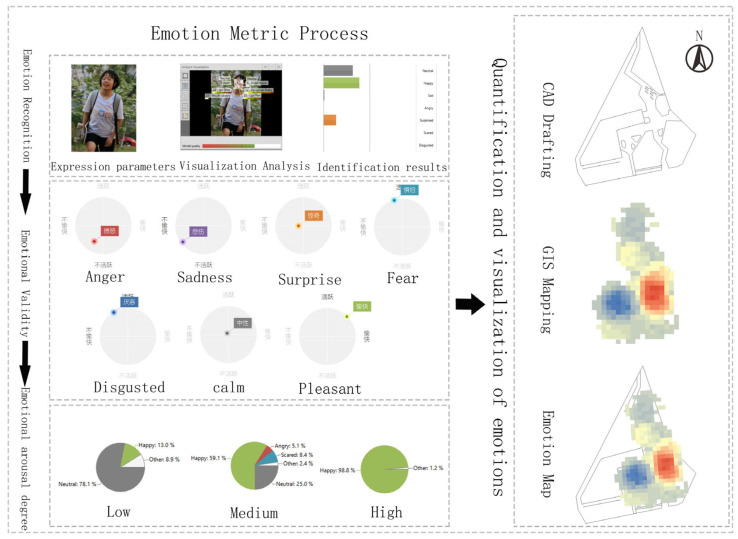
Emotion measurement and emotion map construction process. (Emotion measurement: Measurement of emotions).

**Figure 4 ijerph-20-04277-f004:**
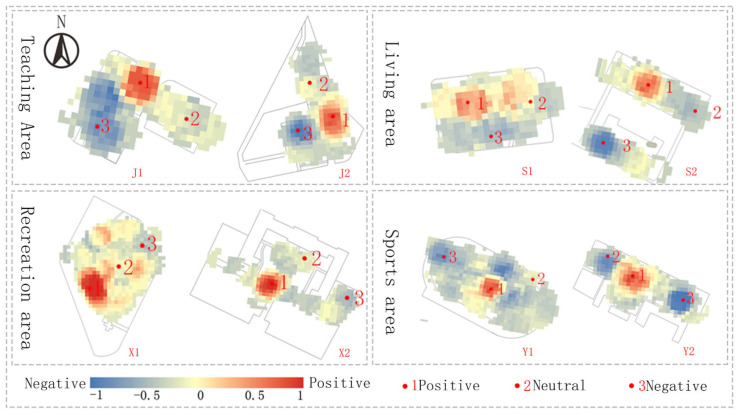
Mood map and mood marker points for each division.

**Figure 5 ijerph-20-04277-f005:**
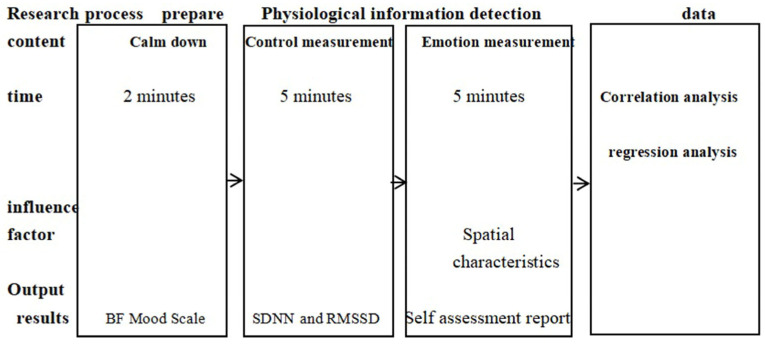
Experimental research process.

**Table 1 ijerph-20-04277-t001:** Descriptive statistics of physiological data.

		Minimum	Maximum	Mean	Standard Deviation
Positive emotion point	SDNN	52.34	106.03	74.01	10.63
	RMSSD	27	74	46.88	9.98
Natural emotion point	SDNN	47.32	101.27	66.48	10.61
	RMSSD	25	62	40.85	8.53
Negative emotional point	SDNN	38.06	79.62	58.59	9.26
	RMSSD	19	53	35.12	7.49

**Table 2 ijerph-20-04277-t002:** Pearson correlation between heart rate variability of spatial attribute elements.

	SV	BP	D/H	SCI	PV
SDNN	0.736 **	0.477 *	0.560 **	0.843 **	−0.759 **
RMSSD	0.814 **	0.485 *	0.561 **	0.842 **	−0.675 **

** Significant correlation at the 0.01 level (two-tailed). * At the 0.05 level (two-tailed), the correlation is significant.

**Table 3 ijerph-20-04277-t003:** Pearson correlation between heart rate variability of landscape amenity elements.

	GV	PTN	NUOSI	NUOSE	NFT
SDNN	0.557 **	0.131	−0.140	−0.027	−0.385
RMSSD	0.485 *	0.087	−0.214	−0.064	−0.330

** Significant correlation at the 0.01 level (two-tailed). * At the 0.05 level (two-tailed), the correlation is significant.

**Table 4 ijerph-20-04277-t004:** Linear regression model of correlation significant spatial features with SDNN.

	Unstandardized		Standardized Coefficient	VIF.	*p*	R^2^	F
	B	Std. Error	Beta				
Constants	63.095	2.922	-	-	0.000 **	0.983 0.975	F = 130.679 *p* = 0.000 **
SV	16.790	3.119	0.270	2.335	0.000 **		
GV	3.534	0.939	0.086	2.094	0.087		
PV	−21.590	2.582	−0.380	1.858	0.000 **		
D/H	0.437	0.130	0.160	2.084	0.004 **		
SCI	0.629	0.078	0.376	2.027	0.000 **		
RL	−2.441	1.620	−0.062	1.585	0.151		
BP	0.606	1.217	0.022	1.794	0.625		

* denotes *p* < 0.05, ** denotes *p* < 0.01.

**Table 5 ijerph-20-04277-t005:** Linear regression model of correlation significant spatial features with RMSSD.

	Unstandardized		Standardized Coefficient	VIF	*p*	R^2^	F
	B	Std. Error	Beta				
Constants	37.304	2.271	-	-	0.000 **	0.982 0.975	F = 122.949 *p* = 0.000 **
SV	19.919	2.425	0.424	2.335	0.000 **		
GV	2.549	1.507	0.083	2.094	0.110		
PV	−10.351	2.007	0.237	1.858	−0.000 **		
D/H	0.336	0.101	0.162	2.084	0.004 **		
SCI	0.449	0.061	0.356	2.027	0.000 **		
RL	−3.645	1.259	−0.123	1.585	0.011		
BP	−0.964	0.946	−0.046	1.794	0.324		

* denotes *p* < 0.05, ** denotes *p* < 0.01.

## Data Availability

The original contributions presented in the study are included in the article; further inquiries can be directed to the corresponding authors.

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
