# Peer review of "The Impact of Campus Outdoor Space Features on Students’ Emotions Based on the Emotion Map"

_ijerph, 2023, doi:10.3390/ijerph20054277_

Round 1
Reviewer 1 Report
Through collecting and processing multiple data and SPSS software, this paper analyzes the different effects of campus outdoor space features on students' emotions. The research method is relatively novel and the workload is large, which has certain practical significance for improving the campus environment construction. However, there are still some problems in the content of the article, which need to be revised before publication:
1. The logic of the introduction needs to be further sorted out, and more updated and authoritative literature should be added, so as to better put forward the research topic and significance.
2.This paper simply analyzes the correlation between variables and lacks in-depth analysis of the research results. It is suggested that the author use more space to analyze the conclusions in more detail so that readers can better understand the research results of the paper.
3. There are some obvious mistakes in the paper, such as the repetition of the figure number, subtitle and content, which is easy to cause confusion. It is suggested to carefully check the whole article to avoid similar mistakes.
Reviewer 2 Report
Thank you for giving me this opportunity to read the manuscript entitled "The Impact of Campus Outdoor Space Features on Students' Emotions Based on the Emotion Map". The topic of this manuscript is interesting and would be a good contribution to this field. I think it could be considered for publication in IJERPH once the following issues are addressed.
1. Please replace the keywords that already appear in the manuscript's title with close synonyms or other keywords, which will also facilitate your paper being searched by potential readers.
2. Appropriate references should be added to support the statement between Lines 26-40.
3. Line 66, “… , and design [16-17]. ” some newly published papers are suggested to cited as references here, for example, “How does urban expansion impact people’s exposure to green environments? A comparative study of 290 Chinese cities”, and “Observed inequality in urban greenspace exposure in China”
4. Line number should be added, which will be easier for reviewers to give comments.
5. Page 2, “impact on human health [13], …”: a paper titled “Dynamic assessment of PM2. 5 exposure and health risk using remote sensing and geo-spatial big data” is suggested to be added as a reference to support the statement here,
6. There is no need to label the abbreviations of some phrases, for example, “sky visibility (SV),”. You just need to label it once at the first time when it appears.
7. Some grammatical errors exist in the manuscript. Therefore, a critical review of the manuscript's language will improve its readability.
Round 2
Reviewer 1 Report
I think the author has made careful revision according to the opinions, and the quality of the paper has been significantly improved. The revised manuscript is acceptable this time.